# Methodology for Designing an Optimal Test Stand for Camera Thermal Drift Measurements and Its Stability Verification

**DOI:** 10.3390/s22249997

**Published:** 2022-12-19

**Authors:** Kohhei Nimura, Marcin Adamczyk

**Affiliations:** Institute of Micromechanic and Photonics, Faculty of Mechatronics, Warsaw University of Technology, 02-525 Warsaw, Poland

**Keywords:** camera, temperature effect, thermal image drift, temperature compensation model, test stand verification

## Abstract

The effects of temperature changes on cameras are realized by observing the drifts of characteristic points in the image plane. Compensation for these effects is crucial to maintain the precision of cameras applied in machine vision systems and those expected to work in environments with varying factors, including temperature changes. Generally, mathematical compensation models are built by measuring the changes in the intrinsic and extrinsic parameters under the temperature effect; however, due to the assumptions of certain factors based on the conditions of the test stand used for the measurements, errors can become apparent. In this paper, test stands for thermal image drift measurements used in other works are assessed, and a methodology to design a test stand, which can measure thermal image drifts while eliminating other external influences on the camera, is proposed. A test stand was built accordingly, and thermal image drift measurements were performed along with a measurement to verify that the test stand did eliminate external influences on the camera. The experiment was performed for various temperatures from 5 °C to 45 5 °C, and as a result, the thermal image drift measured with the designed test stand showed its maximum error of 16% during its most rapid temperature change from 25 °C to 5 °C.

## 1. Introduction

The demand for digital cameras and machine vision systems in industry and research continues to grow. Industrial cameras designed to be capable of working in harsh conditions, such as high temperature, high pressure, and vibrations, are favored in various production lines and research, including medical science and archeological inspections [1]. Specifically designed cameras, such as thermal vision cameras, are also decreasing in price, allowing the expansion of their application to various fields [2]. In medical science, diagnostics based on digital images are now essential for treatments, along with the development of various image processing methods that are region-based or classification-based, which use predefined samples to extract specific features [3,4]. Robotic surgical systems have recently been gaining traction; these rely on high-resolution 2D or 3D medical imaging [5]. Manufacturing industries are increasing their interest in machine vision, as it provides contactless inspection of products and extends the development of production line automation [6]. In the employment of machine vision, the quality of the measurements obtained from image processing is highly dependent on the quality of the images captured by the cameras. Aside from variables such as scene illumination and acquisition parameters, environmental conditions, including temperature, vibration, and pressure, have a substantial influence on the quality of the captured images [7]. In applications that demand high-quality results, such as the ones previously mentioned, recognition of these environmental influences in order to compensate for the errors caused is preferred over trying to control the environment precisely. In particular, in systems that employ multiple cameras to acquire information from a common target from different angles, such as that presented in [6], understanding the environmental influences on each camera is essential to minimize the measurement errors between them.

For most of their applications in industry, vision systems require calibration to compensate for errors due to various optical problems, such as aberrations and distortions [8]. Most calibration processes involve the mathematical description of the geometry of the observed scene and the coordinates of the captured image. The camera subjected to calibration captures images of artifacts containing a predefined geometry and patterns. The geometry in the captured image is measured with pixel-to-mm conversion and compared with the actual geometry. The results are used to obtain the calibration parameters and to develop a mathematical model. The results of the calibration are generally satisfactory; however, due to the assumptions in the mathematical model, its effectiveness may vary for different conditions of the camera. Some of these assumptions come from the environmental conditions of the laboratory where the measurement is conducted to obtain the parameters. Because the models assume the calibration parameters remain unchanged, small errors in image reconstruction can be expected for applications in environments different from the laboratory. For instance, the change in temperature is a major influence on the intrinsic parameters, including the lens geometry and optical distances [9,10].

The effect of temperature changes on digital cameras has been studied to improve their performance in various applications [11,12,13]. In most cases, the effect of the camera warmup has been the main focus of research. When digital cameras are powered on, the electronic components start to produce heat and continue to increase the temperature inside the device’s housing until the camera reaches thermal equilibrium [14]. This increase in temperature usually reaches up to 15 degrees Celsius from the ambient temperature. During this process, the generated heat is transferred to the structural components of the device, causing them to deform according to the thermal expansion of the used materials. As a result, the intrinsic and extrinsic parameters of the camera are changed and can be observed as the thermal image drifts. Image drifts from this process can be from tenths of pixels to several pixels [9,13,15]. An exemplary result of a thermal image drift measurement is shown in Figure 1.

The most versatile method to reduce thermal image drift is to apply a temperature compensation model. In [15], the author presented two possible cases of temperature influence on the geometrical features of the tested camera. The thermal image drift from the camera warmup process was measured to determine the intrinsic parameters, which depended on the presented two cases. The results were used to build a compensation model with a proposed algorithm. The author concluded that the compensation model was applicable to most digital cameras for indoor applications, where a significant change in the ambient temperature is often disregarded. In [16], the authors presented a modified analytical camera model, which had the influence of temperature change implemented mathematically. They assumed that the temperature had no influence on the intrinsic parameters of the camera and conducted an experiment to verify the camera model with the influence of temperature changes on the extrinsic parameters.

In most studies of building a mathematical temperature compensation model, the research is followed by an experiment to verify the model. The experiments are conducted by reproducing the conditions and situations assumed in the model. The frames captured by the camera have the compensation model applied to see whether images without thermal drifts can be achieved or not. In order to obtain reliable results for verification, the design of the experimental setup must be able to mimic the conditions assumed by the model as much as possible and eliminate the factors that are not assumed by the model. For instance, for compensation models that consider the effect on the extrinsic parameters of the camera, the effects of the temperature on all the structural elements between the tested camera and the image artifact should be known. Similarly, if the model considers the change in the dimensions of the image artifact due to varying ambient temperature, its coefficient of thermal expansion and its deformation map in relation to the temperature should be known; if not, the variation in the ambient temperature should be isolated from the image artifact. Any factors that affect the dimensions in the test stand not considered in the compensation model may cause additional image drifts that are not included in the model or may even introduce random errors in the measurements, which hinders the repeatability of the model. Therefore, improper designs of test stands are unlikely to obtain reliable verification results for the compensation model. Even if the test stand eliminates factors that affect its dimensions other than the varying temperature, unrecognized errors or environmental effects can be present to cause unwanted image drifts. Such factors may include the bending of the structure over time, vibrations, errors in material compositions, which deviate the thermal expansion coefficient from the stated value, etc. It is practically impossible to eliminate the factors not present in the mathematical model completely; thus, a verification procedure to confirm the stability of the test stand is essential. This paper focuses on the methods to verify the stability of such test stands.

The remainder of this paper is organized as follows. Section 2 assesses the various methods and test stands for drift measurements described in past works to determine a method for the measurement of the thermal image drift, followed by the requirements of the test stand design optimal for the chosen measurement method. Section 3 describes the test stand design developed according to the requirements described in Section 2. Section 4 describes a method to verify the stability of the thermal image drift test stands. Section 5 analyzes the results of the stability verification, and Section 6 concludes the paper with a discussion of the application of the proposed verification method and its possible future developments.

## 2. Method of Thermal Image Drift Measurements

Generally, detection of the effects of temperature changes on a camera is conducted by extracting the characteristic features from images captured during the camera’s operation under changing environmental conditions. An image artifact with defined geometrical features is placed in the field of view of the camera, and its characteristic points are tracked throughout the measurement process. In this section, the methods to measure thermal image drift measurement from previous research are assessed to determine the requirements for the test stand necessary to obtain reliable measurements of the effects of the ambient temperature changes on an industrial camera.

### 2.1. Test Stand Assessments

#### 2.1.1. Test Stand That Only Assumed Changes in the Intrinsic Parameters

The experiment setup described by Handel in [15] consisted of the tested camera and the image artifact, which was a black metal plate with printed characteristic feature points. The paper did not mention the construction of the setup that defined the rigidity of the positions of the camera and the image artifact; however, it stated that the relative position between the two elements was fixed. This statement was valid since the experiment only tested the temperature variation caused by the camera warmup, and Handel assumed that the extrinsic parameters were not affected. In other words, this experimental setup would be insufficient if we were also interested in the effect of the variation in ambient temperature, which we assume would have an influence on the relative position between the camera and the artifact, depending on the construction built between them.

#### 2.1.2. Test Stand That Employed Less Reliable Structural Materials

In [16], Podbreznik and Potočnik described two experiments, one to test the temperature influence on the extrinsic parameters and another for the intrinsic parameters. For their experiment, to measure the effects on the extrinsic parameters, the camera and the image artifact were placed on each end of a steel structure with a known thermal expansion coefficient, and the ambient temperature was changed between 0 °C and 50 °C. They did not state whether the temperature variation was isolated to selected areas; thus, the image artifact was also likely subjected to the temperature variation. For the experiment dedicated to intrinsic parameters, they designed a structure made of wooden material to maintain a distance of 1 m between the image artifact and the camera. Likewise, the image artifact was subjected to temperature variation. More importantly, despite the fact that this experiment disregarded the influence of the extrinsic parameters, the use of wooden material can be expected to cause randomness in the measurements due to the fact that the dimensions of a wooden structure may depend on environmental factors other than temperature changes, for instance, humidity.

#### 2.1.3. Test Stands That Disregarded the Thermal Influence on the Structural Elements

Pan, Shi, and Lubineau in [12] described their experimental setup to measure the influence of temperature variation on stereo digital image correlation measurements. Their cameras and the image artifact were mounted on a laboratory table, and air conditioning was used to control the ambient temperature. The image artifact was made of quartz glass, and the influence of temperature variation on its dimensions was disregarded due to its thermal expansion coefficient being close to zero. Likewise, the influence on the dimensions of the structure between the elements was disregarded; however, their materials were not listed.

The experimental setup described in [13] is another example of a similar situation. Their setup used a camera to capture frames of an image artifact made of an aluminum plate, and the ambient temperature was controlled using air conditioning. Further details on the construction of the setup were not described. Although the thermal expansion of the aluminum plate was considered in their model, the effects on the other elements within the setup are difficult to determine from the context. Considering that they used air conditioning to control the ambient temperature, the elements present between the camera and the image artifact were likely to have been affected by the temperature variation.

In [9], the authors described an experimental setup to observe the effect of temperature variation on the built-in cameras of a smartphone and a Raspberry Pi. The smartphone was mounted on a carbon tripod and was exposed to varying temperatures caused by its self-warmup while it captured the frames of the designed test target. The Raspberry Pi was fixed on a gauge stand and was exposed to ambient temperature changes using a thermal infrared lamp. The details of the mounts of the devices were not explained; however, if the tripod was only placed on the floor, it would be prone to vibration, and thus the stability of its relative position to the test target would not be ensured.

### 2.2. Conditions for Optimal Thermal Drift Test Stands

Following the observations of the experiment setups mentioned previously, the required conditions for the experiment to collect image drift data with high reliability were finalized as follows:The possibility to change the ambient temperature within a reasonable temperature range that reflects realistic camera operating conditions;The isolation of the temperature variations to only the camera and lens; the used image artifact should be temperature independent, or the positions of its various features in response to temperature changes should be known and calibrated in the camera coordinates;Unchanging positions of the camera and the artifact, irrespective of the temperature changes;Unchanging ambient lighting conditions to properly detect the characteristic points;The possibility of changing the ambient temperature of the camera surroundings is crucial for obtaining image drift data from cameras under varying ambient temperatures; the warming up of the camera is not the only source of a change in temperature. The variation in the ambient temperature should be limited to the surroundings of the tested camera. As mentioned previously, using air conditioning to control the ambient temperature is assumed to have effects on all elements in the testing environment, including the floor and the structures that keep the camera and image artifact stable; thus, it must be avoided for our purposes. The image artifact’s position relative to the camera and its dimensions must be independent of the controlled ambient temperature around the camera. The number of elements that link the camera to the image artifact should be minimized, as well as the types of material used. These conditions are based on our assumptions that the effect of varying temperatures is present in both the intrinsic and extrinsic parameters of the tested camera;Test stands built according to these conditions can be applied to test the effects of temperature changes on various measurement systems which employs a 2D camera, including 3D scanners, microscopes, and various vision systems used for quality control.

## 3. The Test Stand

### 3.1. Test Stand Structure

Based on the requirements of the strategy and what we assessed from the past studies described in the previous section, the test stands to measure the thermal image drift was designed. The finalized test stand consisted of the following major elements:Tested camera IDS UI-5282SE-C Rev.4 [17];Image artifact made from 10 mm thick glass plate and mounted on a rotary–linear stage;Thermal chamber;Invar frame with a shelf to mount the tested camera;Two Akurat S8Mark2 light emitting diode (LED) panels [18].

The camera and the image artifact were mounted on opposite ends of the invar frame for the rigidity of their relative positions. The invar shelf, which was on one of the ends of the frame, allowed the setup to test any camera available on the market. The camera was situated inside the thermal chamber, which isolated the temperature variation to the area surrounding the camera. The image artifact remained outside the thermal chamber, independent of the temperature variation. The thermal chamber was modified with two holes on one of its walls: one as an inspection hole for the camera to capture frames of the image artifact, and the other as a ‘pipe’ with the invar frame running through it to keep the camera and the image artifact mechanically linked. The inspection hole had no glass, as placement of any optical element between the camera and the image artifact should be avoided. Furthermore, if a glass were to be placed, its position would depend on the wall of the thermal chamber, which was not athermalized. In other words, we could not ensure the thermal stability of the thermal chamber walls, which would influence the light rays entering the camera with an optic. Instead, the hole was equipped with an automated flap that could be controlled remotely. This flap could be fully opened permanently or be opened temporarily for the time needed to capture the frame data. The invar frame was mounted on the table outside the thermal chamber, thus creating no rigid links between them. This configuration kept the invar frame unaffected by any deformation of the thermal chamber walls and vibrations caused by the operation of the thermal chamber. Invar was chosen as the material for its low coefficient of thermal expansion to ensure the unchanging mutual positions of the camera and the artifact when subjected to a wide range of temperature changes inside the thermal chamber. The visual of the designed test stand is presented in Figure 2a,b.

### 3.2. Systemization of the Experimental Setup

The experimental setup was stored in a laboratory equipped with air conditioning to maintain the room temperature at 20 °C (stabilization of ±1 °C). The two LED panels were used for uniform scene lighting, and the windows were covered to eliminate parasitic lights. The used thermal chamber allowed the control of the ambient temperature around the camera from −10 °C to 70 °C and was equipped with a communication interface, which was controlled by MATLAB software. Four temperature sensors were placed on different surfaces of the test stand to record the temperature values. The control of the thermal chamber, the frame capture by the camera, and the recording of the temperature values were all conducted remotely, eliminating the necessity for the operator to be present in the laboratory during the measurement process. The test stand was able to automatically conduct measurements, which could last up to dozens of hours or days using predefined commands.

### 3.3. Assumption of the Temperature Influence on the Refractive Index of Air

This design of the test stand introduced the advantage of limiting the temperature variation to only the tested camera. However, this setup exposed the result to influence by the spatially changing refractive indices of air due to maintaining different temperatures inside and outside the thermal chamber. To determine whether this phenomenon would affect the observed image drift, the expected image deformation was simulated considering Edlén’s formula [19], Snell’s law, and the geometry of the experimental setup. The ambient temperature *T_0_* and the ambient temperature outside the thermal chamber *T_0+_*_Δ*T*_ were assumed to be 20 °C and 45 °C, respectively, which corresponded to the maximum temperature deviation between the inside and the outside of the thermal chamber in the conducted measurements. The simplified sketch describing the ray trances in the test stand is presented in Figure 3a.

The refractive indices of air inside and outside the thermal chamber are denoted as $n_{T_{0}+\Delta T}$ and $n_{T_{0}}$, respectively. A single camera ray (marked as a red line) hits the artifact plane at the point *P*(*x*, *y*). The point *P*(*x*′ *y*′) denotes the point *P* coordinates for the case of no temperature difference $T_{0}+\Delta T=T_{0}$ in which the refractive indices are equal as $n_{T_{0}+\Delta T}=n_{T_{0}}$. Using the exact geometric dimensions of the test stand as well as Edlén’s formula and Snell’s law, the expected image deformations caused by the changing refractive index of air due to the differences in the temperature inside and outside the thermal chamber were simulated. The output of the simulation is presented in Figure 3b. The maximum image drift in the image artifact plane, caused by the temperature increment of 25 °C, was no greater than 2.5 µm at the corners of the image, which was small enough to be negligible compared to the observed image drifts.

## 4. Verification Method

Following the assembly of the test stand described in the previous section, a set of measurements was conducted to verify its stability. This section describes the experimental setup and the procedures followed in conducting the measurements.

### 4.1. Test Stand Extension

As mentioned previously in this paper, due to the time required for thermal stabilization after each change in the temperature inside the thermal chamber, the thermal drift measurements may last up to dozens of hours. The stability of the test stand can be ensured if we can confirm that any environmental or mechanical effects have negligible influence on the dimensions of the test stand over a long measurement time and that the temperature variation of the camera is the only significant influence on the observed image drifts. In the case of our test stand, our main target of this verification was to determine whether the thermal expansion effect on the invar frame had a negligible influence on the drift measurement. This measurement also helped us to confirm whether the provided invar material had properties accurate to its specification. The verification setup was conducted by a simple modification to the prepared test stand. An additional camera, Flir GS2-GE-50S5M-C [20], was mounted on a segment of the invar frame outside of the thermal chamber to observe the tested camera during the temperature-varying process. The photo of the test stand with the additional camera is shown in Figure 4.

Four circular markers with diameters of 5 mm were attached to the invar camera stage, visible to the observing camera. Figure 5 shows the sample frame with the tested camera and the markers taken by the observing camera.

### 4.2. Experiment Process

The tested camera was exposed to a series of temperatures inside the thermal chamber, replicating the conditions during the thermal drift measurement. The temperature settings were applied in the following order: 25 °C, 35 °C, 25 °C, 15 °C, 25 °C, 45 °C, 25 °C, 5 °C, and 25 °C (graphically presented in Figure 6). For each of the nine temperature steps, the camera was exposed to the set temperature for 90 min to ensure that it would reach its thermal equilibrium state. The total sequence took about 13.5 h, and during that time, the tested camera captured frames of the image artifact with the same procedure executed for the drift measurements. Simultaneously, the observing camera captured a frame every 30 s, recording a total of 1619 frames of the tested camera and the stage with the markers. The ambient temperature was recorded during the measurement inside and outside the thermal chamber. The LED panels faced the tested camera for improved lighting inside the thermal chamber. From the captured frames from both cameras, the centers of each marker were calculated, and the drifts were measured as the shift of these centers on the image plane.

The main advantage of this verification method was that it could be conducted simultaneously with the thermal drift measurements. The main scope of this verification was to detect the shift in the camera position with respect to the image artifact position and confirm that the temperature change was the only significant influence on the image drifts. By recording the two datasets at the same time, the correlation of the camera position and the image drift could be determined for each specific time a frame was recorded. Other methods, such as employing multiple proximity sensors to detect the camera position, may provide results with higher precision; however, because the sensors must avoid the temperature change influence, using them within the limitations of the distance from the target object would require further modifications of the test stand. The proposed method could detect position shifts in two dimensions with the use of just one camera placed under the same environmental conditions as the image artifact.

## 5. Results

After the frames from the verification experiment were collected, the drift calculation of each marker was conducted separately using a subpixel algorithm implemented in MATLAB, which was resistive to thermal noise and was able to calculate valid results for various ambient lighting and marker sizes [21,22,23]. The data collected from the two measurements conducted in parallel were analyzed for their correlation. From the frames of the stability observation, the maximum shift of the camera position caused by the thermal expansion of the invar frame was obtained. Assuming that the shift had an influence on the thermal image drift, the amount of drift caused by the shift of the camera position can be obtained. The result of the drift calculation of a single marker attached to the invar shelf is presented in Figure 7. What is important to point out is the intense peak that was visible at the beginning of the final temperature step. At this point of the measurement sequence, the inside of the thermal chamber went through a rapid temperature increase from 5 °C to 25 °C. This caused water droplets to condense on the surfaces of the invar frame, applying noise to the captured frames, and as a result, the accuracy of the center track was decreased. These intense peaks were unrelated to the mechanical stability of the test stand; thus, they can be considered outliers. Excluding the outliers, the plots showed high stability of the marker center throughout the long temperature variation sequence. Considering the center position during the starting temperature of 25 °C as a reference, the maximum drift measured was less than 0.3 pixels for both directions. Converting the pixels from the frames to millimeters, which was around 30 µm, and this confirmed the high stability of the invar frame under an intense temperature variation.

The registered thermal image drift captured by the camera inside the thermal chamber is shown in Figure 8. From the thermal drift measurement, we confirmed a maximum drift of 1.49 pixels, calculated to be about 0.18 mm. Assuming that the camera position was shifted by 30 µm, the expected drift caused would be about 0.24 pixels. This was about 16% of the maximum drift measured and less than the average drift observed from all markers. Assuming that the largest drift measured from the two cameras occurred during the same temperature step, the actual maximum thermal image drift caused purely by the temperature effects on the tested camera can be said to be about 16% less than the maximum drift measured using this test stand. This was for the maximum drift observed during the temperature step of 5 °C; therefore, the actual momentary drifts during the other temperature steps would have even less difference from the measured drifts. Considering that the drift of the markers on the camera stage was consistent after excluding the outliers, its mean value can be used for the corrections of the drifts in the other temperature steps. The measured mean drift of the camera stage marker, 0.074 pixels, was converted to 7.4 µm, and the shift appeared as 0.05 pixels in the thermal image drift measurement, which had no greater significance than noise.

## 6. Conclusions

This paper presented a verification method for the stability of experimental test stands for thermal image drift measurements. After assessing the test stands presented in other works, a methodology to design an optimal experimental setup was performed. Depending on the target information to be collected and assumptions made for the compensation models, the design of the experiment environment and the choice of test stand material can be crucial for obtaining accurate results. Verification of the stability of the test stand is crucial to confirm the accuracy of the thermal image drift measurement, as this is what defines the compensation models used by various industries. According to the determined requirements, the experimental setup was designed and built, followed by a set of thermal image drift measurements. Then, measurements to verify the stability of the test stand were performed. We concluded that the effect of the temperature variation on the invar test stand was negligible for the thermal drift measurement, and the protection of the test stand from unwanted environmental effects was confirmed.

This verification method should be applicable to confirming the stability of all test stands for thermal drift measurements, leading to the ability to confirm the accuracy of the corresponding thermal compensation models. The isolation of the temperature variation will be the most important requirement, as the observing camera must not be exposed to temperature changes. Consideration of a verification method can lead to further optimization of the test stand design for future research on industrial cameras. For future work, the verification method can be extended to verify translations in the third axis. Additionally, methods to detect more detailed effects on the positions and the geometry of the camera lens can help correlate the effect of the lens to the trends of the measured thermal image drift.

## Figures and Tables

**Figure 1 sensors-22-09997-f001:**
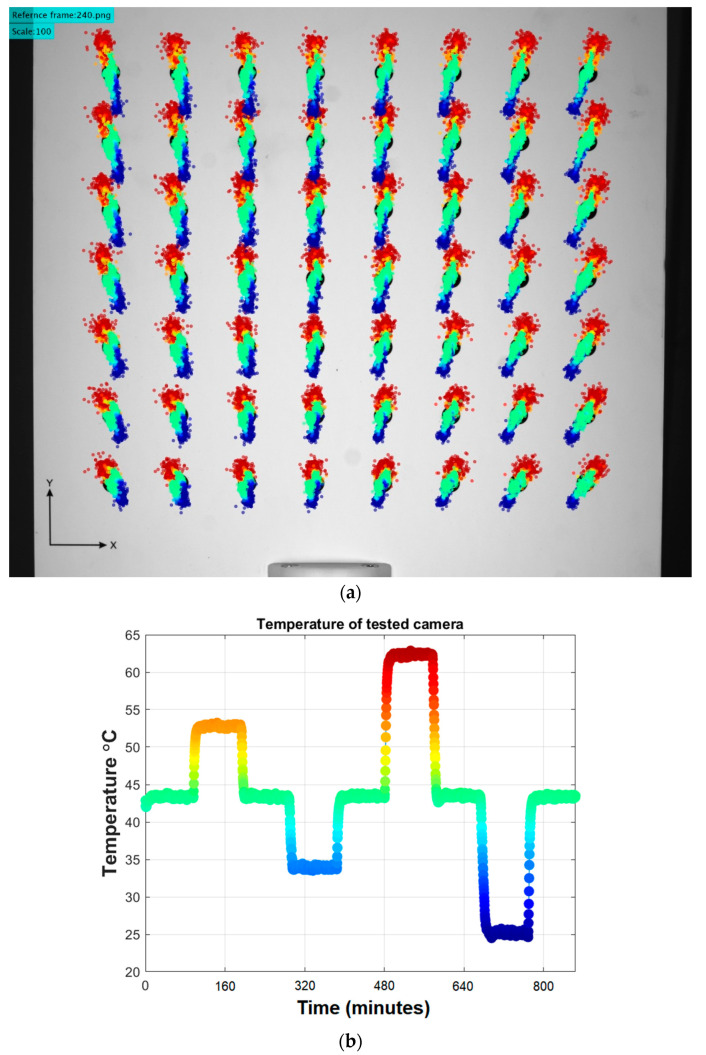
(**a**) Example of a registered thermal image drift with respect to the initial positions of 56 markers forming arrays. All marker trajectories are scaled ×100 for better visualization. The color of each marker represents the temperature of the camera during its frame capture. (**b**) Plot representing the temperature values of the camera during each frame capture. The color scale is consistent with the drifts shown in (**a**).

**Figure 2 sensors-22-09997-f002:**
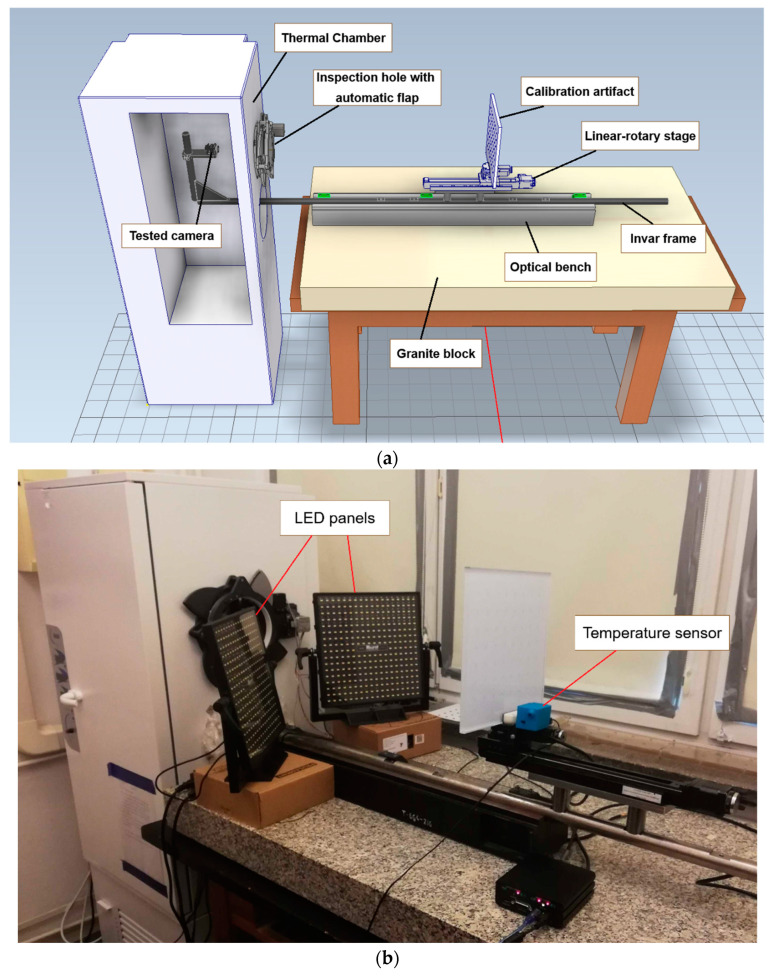
Designed test stand. (**a**) The schematic view of the test stand with the thermal chamber and the invar frame. The invar frame is clamped onto an optical bench, which is stationed on a granite block. A linear–rotary table is mounted on the invar frame, which can control the position of the calibration artifact along the optical axis. The thermal chamber has an opening for the invar frame to pass through and another opening for inspection purposes, which has a controllable automatic flap mounted. (**b**) Photo of the actual test stand stored in the laboratory. Here, the LED panels and the temperature sensor to record the room temperature are visible.

**Figure 3 sensors-22-09997-f003:**
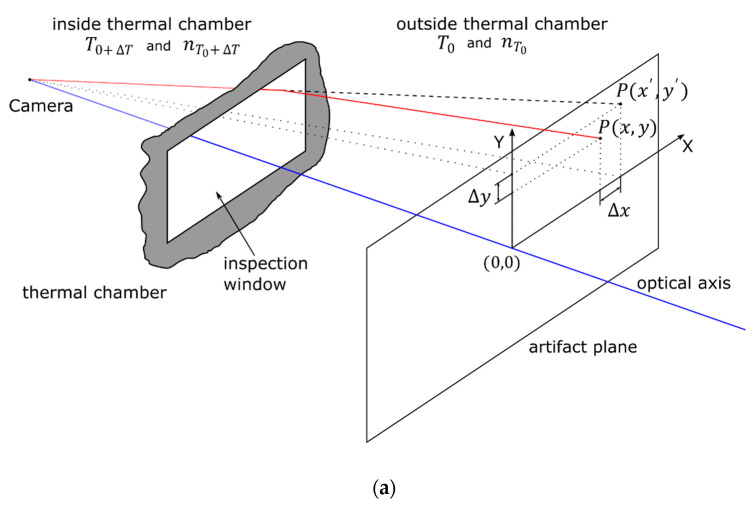
(**a**) The geometry of the test stand used for simulation. The blue line represents the optical axis of the tested camera. The red camera ray is refracted while leaving the thermal chamber due to the changes in the refractive index of air caused by the temperature difference. (**b**) The simulated deformations in the calibration artifact plane caused by the increase in temperature of +25 °C.

**Figure 4 sensors-22-09997-f004:**
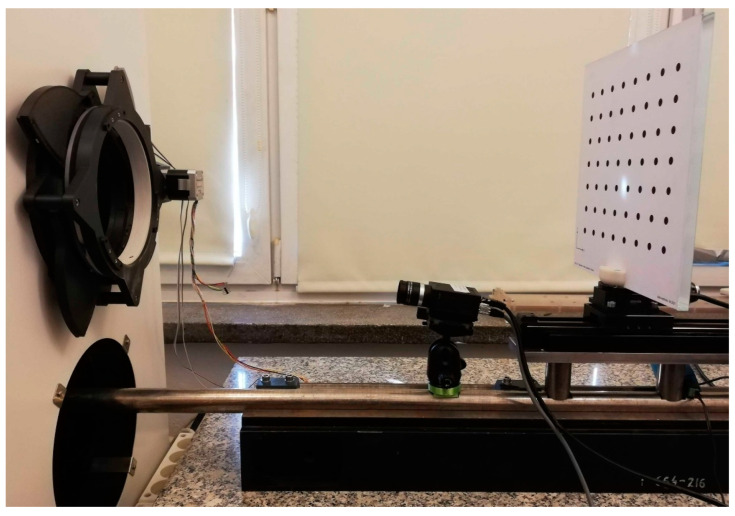
Photo of the test stand with the additional camera attached for the stability measurement. The light-emitting diode (LED) panels were temporally removed for a better view of camera.

**Figure 5 sensors-22-09997-f005:**
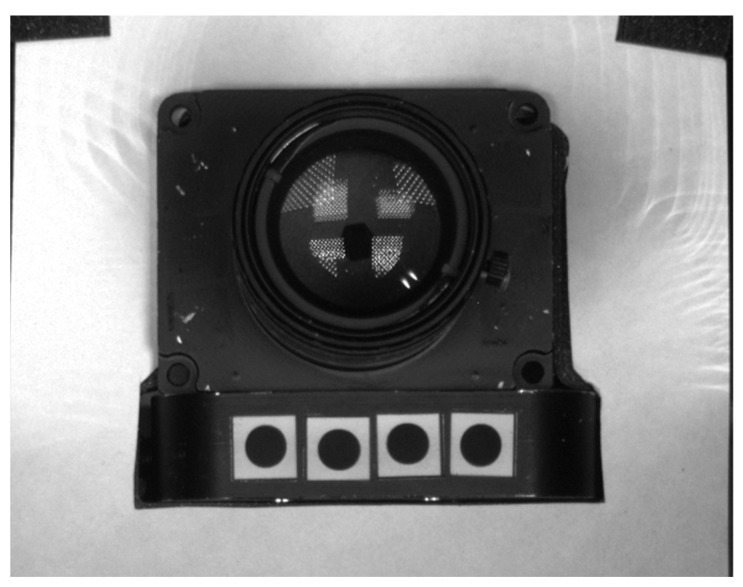
Scene of the tested camera and the markers on the surface of the invar camera stage observed from the observing camera, which is positioned outside the thermal chamber.

**Figure 6 sensors-22-09997-f006:**
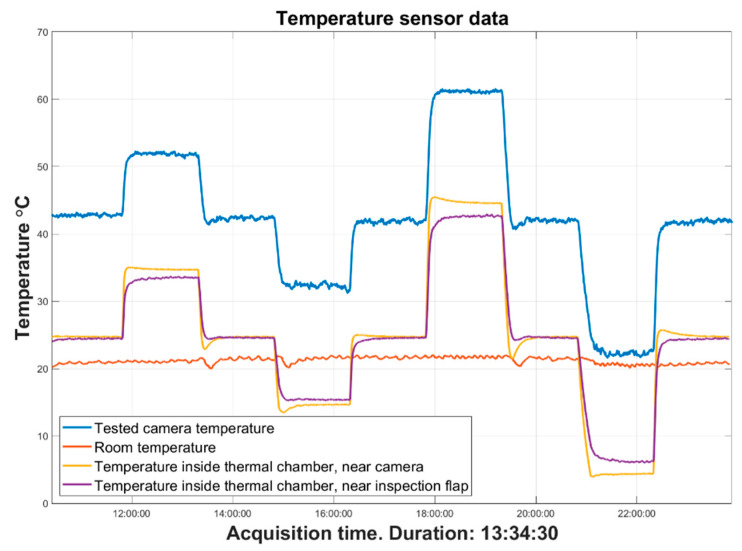
Temperature data recorded during the stability measurement. The blue line refers to the temperature of the observed camera inside the thermal chamber, recorded by a sensor inside the camera; the orange line refers to the room temperature of the lab recorded by the sensor positioned near the image artifact; the yellow line refers to the temperature inside the thermal chamber, recorded by the sensor positioned near the observed camera; the purple line refers to the temperature inside the thermal chamber recorded by the sensor positioned near the inspection flap.

**Figure 7 sensors-22-09997-f007:**
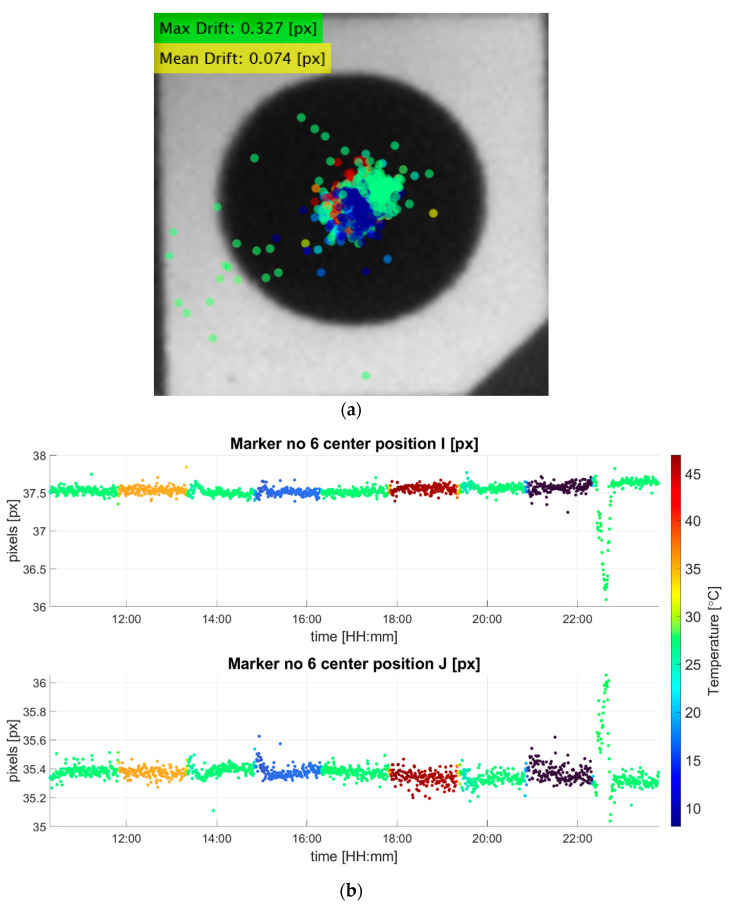
(**a**) The registered center drift of marker 6 (the rightmost of the four markers on the camera stage). The presented maximum and the average drift distance were obtained after excluding the outliers and are in units of pixels. (**b**) The graphical representation of the drifts in the horizontal (I coordinate) and vertical (J coordinate) of the image plane measured in pixels. The color of the points represents the ambient temperature inside the thermal chamber during the measurement.

**Figure 8 sensors-22-09997-f008:**
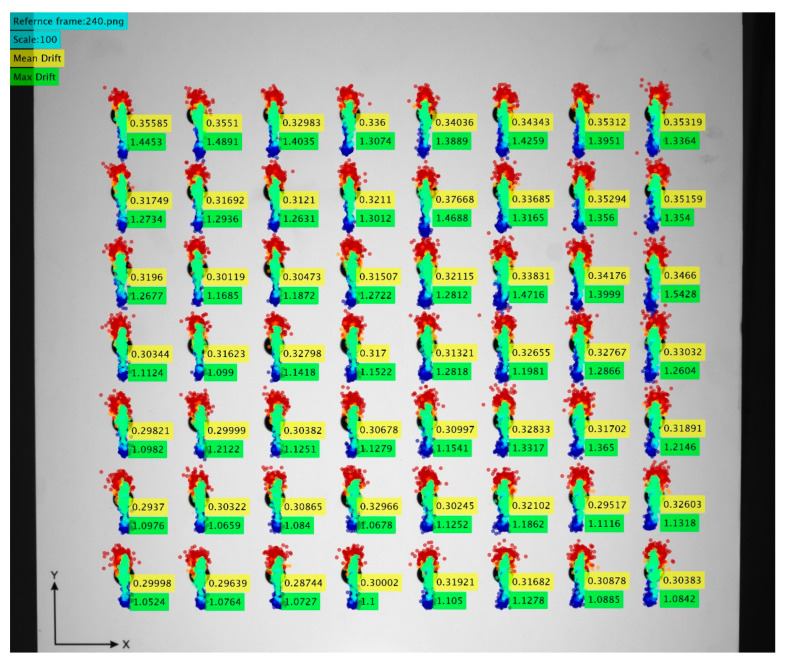
The registered thermal image drift recorded by the tested camera inside the thermal chamber during the parallel measurement with the stability observation. The mean and the maximum drift value for each marker are shown in pixel values. The drifts are scaled ×100 for better visualization. The color of the markers represents the temperature of the camera during the frame capture and is consistent with the temperature scale shown in Figure 7a.

## Data Availability

Not applicable.

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
