# Peer review of "Methodology for Designing an Optimal Test Stand for Camera Thermal Drift Measurements and Its Stability Verification"

_sensors, 2022, doi:10.3390/s22249997_

Round 1
Reviewer 1 Report
The method of the paper has some practical value.The problems in the paper are as follows:
1.The paragraph from lines 145 to 190 is too long. It is suggested that the author divide it into sections and levels. There are several similar paragraphs in the article, and the author is suggested to revise them together.
2.What are the advantages of this method over other methods? The author is suggested to add related content.
Reviewer 2 Report
The research is good enough to advocate the design and verification stability however there are few points which need to be more clear such as
· Title is not justified
· There is application mentioned except the optimal research used
· Why temperature profile was necessary? And what is meant by ‘frames’ in figure.1
· Literature is almost zero
· The mathematical code or simulation methods should be mention
· Conclusion is too long.
Reviewer 3 Report
Manuscript: sensors-2067366
The manuscript titled "Methodology of designing an optimal test stand for camera thermal drift measurements and its stability verification" by Nimura and Adamczyk present results on the use of a different methodology to correct drift observed during temperature change. I have following observations:
1// The temperature change under consideration is 25degC and 5degC. However, can they also predict the effect when the temperature varies to 45degC?
2// Authors have used an optical bench and a thermal chamber. How practical this system shall be for some specialized applications?
3// In figure 6, the change in the temperature sensor data is much more when compared to the initial stage of the acquisition time. Please also include further data to see the impact upon prolonged usage.
4// Please also include other results reported so far in the final section (section 5) of the manuscript to compare them with the existing data.
Based on these, I recommend a Minor revision for the present manuscript.
Reviewer 4 Report
In this paper, test stands for thermal image drift measurements used in other works are assessed and a methodology to design a test stand which can measure thermal image drifts while eliminating other external influences on the camera is proposed. A test stand was built accordingly, and thermal image drift measurements were performed along with a measurement to verify that the test stand does eliminate external influences on the camera. As a result, the thermal image drift measured with the designed test stand showed its maximum error of 16% during its most rapid temperature change from 25°C to 5°C. Generally, this is a good work. It can be accepted if the authors can consider the following issues: 1. Compared to the existing work, what is the main advantage of the proposed method? 2. Why did the authors select the temperature changing from 25°C to 5°C? How about the case temperature changing from 5°C to 25°C? 3. More related works on the object detection are welcome to enrich the literature review such as Transnational image object detection datasets from nighttime driving; Improved Vehicle LiDAR Calibration With Trajectory-Based Hand-Eye Method 4. More words are welcome for the figures.
